# Elastodontic Appliances for the Interception of Malocclusion in Children: A Systematic Narrative Hybrid Review

**DOI:** 10.3390/children10111821

**Published:** 2023-11-17

**Authors:** Vincenzo Ronsivalle, Ludovica Nucci, Nicolò Bua, Giuseppe Palazzo, Salvatore La Rosa

**Affiliations:** 1Department of Medical-Surgical Specialties—Section of Orthodontics, School of Dentistry, University of Catania, Policlinico Universitario “G. Rodolico-San Marco”, 95123 Catania, Italy; nbua@gmail.it (N.B.); gpalazzo@unict.it (G.P.); salvo.larosa11@live.it (S.L.R.); 2Department of Medical-Surgical and Dental Specialties, University of Campania Luigi Vanvitelli, 80100 Naples, Italy

**Keywords:** elastodontics, interceptive ortodontics, pediatric dentistry, narrative review

## Abstract

Background. Interceptive orthodontic treatment aims to eliminate factors that prevent the harmonious development of the maxillary and mandibular arches during childhood, and elastodontic appliances (EAs) represent a group of devices with an increasingly important role. This systematic narrative hybrid review (HR) aims to provide an overview of the clinical indications for the use of EAs according to the available evidence and to identify potential research areas for unexplored applications. Materials and methods. To assess the available literature on the subject, selective database searches were performed between July 2023 and September 2023. With the assistance of a health sciences librarian, a search strategy that utilized terms related to elastodontic therapy was developed. Embase, Scopus, PubMed, and Web of Science were the databases used. Results. The current literature addressing the usability of EAs is scarce and mostly limited to case reports and case series. After 2168 citations were found through the searches, 13 studies were ultimately included. In this regard, information about the clinical use and effectiveness of EAs are reported in a narrative form, defining specific domains of the application that are clinically oriented, including sagittal and transversal discrepancies, atypical swallowing, teeth malposition, two-phase orthodontics and a lack of teeth retention. Conclusions. Within the intrinsic quality limitation of the available literature, it seems that EAs may represent a promising treatment alternative for managing mild-to-moderate malocclusion in children as an adjuvant therapy to the interruption of spoiled habits.

## 1. Introduction

Malocclusion is defined as any alteration of the physiological relationship between dental arches with or without an irregularity of the teeth [1] and represents a condition with a high prevalence and treatment needs [2]. Some types of malocclusions are self-correcting with age, especially if they are not caused by bad habits (such as atypical swallowing or thumb sucking, or mouth breathing) or skeletal disorders that may compromise the function of the oral cavity. For the remaining part of malocclusions that do not self-correct with aging, childhood orthodontic therapy with elastodontics in early mixed dentition has been shown to be a useful tool as an interceptive treatment [3,4,5]. Otherwise, advanced surgical interventions such as orthognathic and/or aesthetic hard or soft tissue surgery are needed to ensure the functional balance between components such as the forehead, eyes, chin, nose, and lips, which are also important for facial attractiveness in the late period [6].

Although there is evidence that early treatment may result in a stable occlusion [7,8], other studies suggest that children do not benefit from interceptive intervention other than a transient improvement in self-esteem [9,10]. As a result, there is no solid consensus about the real effectiveness of the interceptive treatment. In particular, interceptive treatment may be required when it is important to eliminate factors that are preventing the harmonious development of the maxillary and mandibular arches. These aspects frequently cause the skeletal and dentoalveolar systems to react in a compensatory way to maintain stable function and occlusion.

The use of functional devices expanded throughout Europe in the 20th century before spreading to numerous other nations [11,12,13,14,15,16,17,18]. The main goal of functional treatment is to “guide” the proper formation of the bone bases by activating the perioral muscles. Functional appliances are a class of active or passive devices that exert stresses on the orofacial musculature and transmit them to the teeth and jaws, causing changes to the teeth’s alignment and structural integrity [19]. According to certain reports, 15% of mixed dentition disorders can be corrected, and 49% of them can be improved with interceptive orthodontic treatment [20]. The majority of conventional functional devices used in interceptive treatments are created with rigid and semi-rigid structures [21].

With the development of synthetic, elastic, and biocompatible materials, there has been a spread of new appliances called elastomeric devices [21].

Elastodontics is an interceptive treatment that uses removable silicone elastomer appliances to promote the growth of small, biological elastic forces [22]. With the help of these pressures, malocclusions that are present at a developmental stage can be corrected, possibly impacting growth [22,23,24]. While the vestibular flanges stop the perioral muscles from influencing the movement of the teeth, the elastomeric material and the neuromuscular system work together to enhance orthodontic movement [22,25].

Elastodontic devices are manufactured to solve orthodontic problems in mixed dentition, such as the molar relationship on the sagittal plane of class II subjects, crowding, overjet, overbite, crowding, small rotations, and mild class III subjects or ones that are pseudo third class. They can also be used as retainer devices in permanent dentition.

Although the interest in and use of EAs (elastodontic appliances) is increasing among researchers and clinicians, the available literature supporting the clinical applications of these devices appears to be poor and addresses different clinical topics.

Literature reviews can be conducted in a variety of ways, such as “narrative,” “systematic,” and “scoping” approaches. While scoping reviews are a new tool for mapping the body of research on particular arguments, the two primary techniques for gathering and analyzing the literature are narrative reviews (NRs) and systematic reviews (SRs), each one with advantages and disadvantages of their own. NRs offer a comprehensive synopsis of the body of literature already in existence, but they omit a thorough explanation of the methods used to find pertinent data. As an alternative, SRs use a thorough search approach and clearly defined inclusion and exclusion criteria that can be repeated to produce results that are comparable to one another in order to prevent bias in the search and selection of the literature.

Data synthesis and interpretation in SRs can be handled using a narrative approach or meta-analysis (quantitative data), depending on the data findings from the studies that are available. In both cases, SRs offer the best quality of evidence on a particular subject that has been sifted through the search strategy, but they are unable to adequately address clinical questions that necessitate a thorough understanding of the phenomenon.

A hybrid approach to literature reviews has recently been proposed. It incorporates elements of both systematic and narrative approaches, with a focus on rigorous methodology for study selection and search strategies, combined with a less constrained and narrative description of the data findings [26]. This systematic narrative hybrid review (HR) aims to provide helpful information about the factors that should be taken into account, from diagnosis to clinical applications, in the management of patients that can be treated with elastodontic devices and to provide an overview of the literature that is available regarding this treatment approach. In order to achieve this, we divided the results into distinct domains and investigated the possibility that any domains were underrepresented.

## 2. Materials and Methods

The structure of the current manuscript adheres to the essential components of the suggested methodology for conducting HR [26].

### 2.1. Research Question

In order to inform the parameters of the search strategy, we defined a primary research question, the answers to which could be used as a guide by orthodontists: “Which malocclusions and/or dental developmental discrepancies may be amenable to treatment with elastodontic devices”?

### 2.2. Justification

The lack of a thorough and comprehensive overview in the literature that addresses the actual evidence-based medical and dental indications for elastodontic treatment, as well as the factors that dental specialists should take into account when treating children with malocclusions, served as the rationale for conducting the current hybrid NR. The consolidated methodology for SRs [27] served as the basis for the search protocols and inclusion/exclusion criteria for this purpose, and a narrative approach was employed to analyze the articles that made the shortlist [26].

### 2.3. Eligibility Criteria

All the studies concerning elastodontic treatment in interceptive orthodontics were considered, excluding reviews. There were no limitations to the publication year or language. The studies were also included when they exhibited the following characteristics reported according to the PICO format: studies conducted in growing human subjects (participants); studies evaluating treatment effects using elastodontic devices (intervention), before and after treatment outcomes or considerations with elastodontic appliances (comparison), studies assessing occlusal radiographical data (dental and/or skeletal outcomes) (Outcomes).

### 2.4. Literature Sources and Search Parameters

Selected database searches were conducted up until September 2023 in order to evaluate the body of literature currently available on this topic. A health sciences librarian helped create a search strategy that made use of all found keywords and free-standing terms. The databases that were used were Web of Science, Embase, Scopus, and PubMed. Supplementary research was examined to ensure that all evidence sources listed in the reference list were valid. Table 1 presents the results of changing the approach search for each database.

### 2.5. Data Cleaning

#### 2.5.1. Study Selection

After the search results were obtained from all electronic databases, the citations were loaded into the reference manager application EndNote X9 (Clarivate^TM^, London, UK). Duplicate reports were removed, and articles offering updates or interim results were only evaluated once. Before proceeding through the complete texts of any relevant studies, two writers named V.R. and S.L.R. verified all of the titles and abstracts that they had gathered from the databases. The eligibility of the studies was assessed impartially, and any discrepancies were resolved following consultation with an additional author, G.P. The level of agreement between the two reviewers was evaluated using Cohen kappa statistics.

#### 2.5.2. Data Extraction

Two authors (S.L.R. and V.R.) developed a data extraction form to collect the attributes and results (study design, sample size, and objectives) required for the subsequent literature analysis. We talked about any differences with an additional author reviewer (G.P.). Cohen kappa statistics were used to assess the level of agreement between both writers.

### 2.6. Information Synthesis

The proposed method for HRs [26] required a narrative presentation of the findings from the chosen papers. The results were reported using a methodology that was derived from earlier studies and which other researchers had published. Results were arranged and discussed to create distinct domains that encompassed all data retrieved from the included studies in order to more effectively address clinical indications.

## 3. Results

The reviewers examined 2168 records out of the 2403 citations that the strategy searches turned up after removing duplicate files. After the titles and abstracts were read, a total of 2029 articles were removed, and the full texts of 120 articles were then obtained. After a comprehensive examination of the entire texts of these papers, 13 studies were judged to be appropriate for the review. Three studies were case reports [28,29,30], three were case–control studies [31,32,33], one was a case series [34], five were retrospective studies [21,35,36,37,38], and one was a prospective study [22].

Appendix A displays the features of the included studies, and Appendix A lists the publications that were removed at this time, along with the explanations for their removal. A summary of all data related to the study selection is given in Figure 1.

The information found was proposed in a narrative review format, creating the domains to facilitate the understanding of the clinician (sagittal discrepancies, transversal discrepancies, atypical swallowing, bad habits, two-phase orthodontics, and lack of teeth retention). Also, the discussion of data findings was also supported by the iconographic representation of clinical examples for the use of elastodontic appliances in children. Figures were retrieved from the clinical database of the Department of Orthodontics of the University of Catania.

### 3.1. Sagittal Discrepancies

Two studies [34,38], one retrospective and one case series, reported the elastodontic efficacy of interceptive treatment with elastodontics in mild class III patients. Instead of the expected simple dentoalveolar changes, the use of EAs favored the anteroposterior relationship between the maxillary and mandibular arches with significant skeletal changes (the ANB angle, which is the angle formed by points A, nasion, and B in lateral cephalograms). But, the authors stress that the decrease in the SNB angle—the angle between point sella, nasion, and point B on lateral cephalograms—was primarily responsible for the skeletal changes. This decrease was impacted by the mandibular postural changes following the correction of the anterior cross bite.

In class II patients, SNA values (detected between point sella, nasion, and point A on lateral cephalograms) measured one year before and after treatment with EAs did not significantly differ from one another, but ANB values were reduced to physiological measurements, and SNB recordings showed a significant increase before and after treatment according to a prospective study and a case series [22,34]. The efficacy of EAs was also important for class II patients with high overjet values in a prospective study [22].

### 3.2. Transversal Discrepancies

Functional posterior cross bites can be treated with elastodontic devices. The inter-molar width (IMW) and inter-canine width (ICW) significantly increased in patients with this device in their mouth in a recent prospective study [36]. Various studies in the order of one retrospective, one prospective, one case series, and two case reports [21,22,28,29,34], have reported that early anterior crowding issues are addressed by EAs through their design, such as the creation of an elastic channel in the anterior teeth, which promotes the dentoalveolar process’ transversal growth [39].

### 3.3. Atypical Swallowing, Spoiled Habits, and Teeth Malposition

EAs not only promote muscle growth but also prevent bilabial contact, compel the tongue to assume the correct position and instruct on proper swallowing techniques, according to a retrospective study [37]. By eliminating functional deficiencies in the stomatognathic system and operating in three dimensions within the oral cavity, elastodontics improves postural problems, bad habits, and breathing (which can occasionally be incorrect in class II patients with a tendency to oral breathing) according to a case–control study [33].

### 3.4. Conventional Two-Phase Orthodontics

According to a case report [28], EAs allow for the treatment of an extractive case in an interceptive phase, which is followed by fixed therapy. After four months of active treatment with EA and myofunctional therapy, the authors showed that in a thirteen-year-old patient with class I malocclusion and severe crowding in both arches on a skeletal class II base, as well as severe crowding in the maxilla and mandibular arches, an augmented overjet (4 mm) and overbite (4.5 mm), a deep curve of Spee and a shift in the upper midline to the right, that deep overbite was diminished to an edge-to-edge relationship and the anterior teeth were properly aligned, with some mesial tipping of the first molars.

### 3.5. Lack of Teeth Retention

Thanks to the minimum anchorage of the teeth, a case report [29] reported the efficacy of EAs in subjects affected by conditions such as dentinogenesis imperfecta, which does not permit the use of conventional orthodontic devices.

## 4. Discussion

Elastodontic devices are a new possibility of treatment in interceptive orthodontics, and the small number of studies already published are evaluated in this hybrid review. To facilitate the use of these appliances by clinicians, in this section, the article is divided into specific domains (the history of EAs, design principles, treatment protocol and description, sagittal discrepancies, transversal discrepancies, atypical swallowing, bad habits, two-phase orthodontics, lack of teeth retention and limitations of the study).

### 4.1. History of EAs: From the “Positioner Appliance” to the Recent Application of Elastic Materials in Orthodontics

In 1945 Kesling presented the forerunner of the elastodontic appliance, the “positioner”, a single elastomeric device with intercuspation for the lower and upper teeth in normal occlusion. He created it as a finishing device or as retention following multibrackets therapy that did not ask for impressions or setup creation. Following orthodontic treatment, the positioners’ elastomeric substance typically permits slight tooth movement [40,41].

In 1950, two orthodontists named Soulet and Besombes created the first functional elastomeric appliance known as an “activator” in France. The activator’s material allowed for the induction of neuromuscular and skeletal effects, as well as the renormalization of the mandible and maxilla positions within the entire cranial system. This treatment was classified as interceptive orthopedics by Besombes [42,43].

To cure malocclusions, Bergersen created a prefabricated elastomeric appliance in 1975. The Occlus-O-Guide^®^ or eruption guidance appliance (EGA) (ORTHO-TAIN, Inc., Bayamon Gardens, Puerto Rico) exhibited the features of a functioning appliance and a positioner combined. It was created with a slight positional correction or dental eruption guidance in mind [22]. The mandibular advancement to the correct class II sagittal discrepancies and a vertical aperture in the front region allowed for the increased vertical development of the posterior teeth and were characteristics shared with functional devices [25,44,45]. Elastodontic appliances do not harm oral soft or hard tissues when in use because of their soft substance [46].

### 4.2. Design Principles of EAs

All these devices present vestibular and lingual flanges; the teeth are placed in a central area that can have indentations, act as a positioner, or have free central space to avoid teeth constriction or the development of orthodontic movement [47]. Up to the border, the vestibular flanges operate as a sort of lip bumper, and the fornixes also stimulate the matrix bone proprioceptively. As a result, both arches are active at the same time, performing an orthopedic action in the following four different planes: vertical, transverse, sagittal, and torsional [43].

As a functional appliance, the upper and lower planes can be collocated at a particular position to encourage or inhibit mandibular advancement [48]. To manage verticality and encourage or inhibit the eruption of the posterior teeth, the occlusal plane might be flat or thicker in the anterior or posterior regions. In cases of atypical swallowing, there is a supplementary ramp and a button on the lingual flange, which is crucial for functional rehabilitation as it serves to guide the tongue onto the palate.

To provide light, biologically elastic stresses without damaging the oral mucosa, soft silicone elastomers or polymer/elastomer combinations are used as construction materials [21], but there are different levels of hardness for EAs that the clinician can choose.

Thanks to their design, the dental–craniofacial and cervical–postural systems, as well as the neuromuscular system, alveoli, TMJs (temporomandibular joint system), soft tissues, salivary glands, and mandibular and maxillary bases, are all reflected in the functional activity of EAs [21,25,42,49,50,51,52,53,54].

The occlusion, along with the tongue, masticatory muscle, lips, and cheeks, interact with the material of elastodontic appliances to produce an ideal space known as the “elastodontic space” within which the movement of teeth takes place. As a result, the device itself does not control dental movement. This neutral zone in which they are positioned is created by the harmony between the muscles of the lips, tongue, and cheeks [21,25,28,37,42,49,50,51,52,53,54,55].

The correct size of an elastodontic can be chosen through a wax bite and the indicators present in the appliance.

Clinical evidence shows that elastodontic devices are adequately tolerated by patients because of their structure and elastomeric material. They are also cheaper than functional ones because they are prefabricated. From this point of view, elastodontics enable the clinician to avoid impression maneuver, which is not comfortable for the patient because no construction bite is required. Moreover, although there is not much available in the way of research, EAs are not susceptible to caries, and they do not harm the periodontium or promote root resorption [43]. From this perspective, it is important to emphasize that elastodontic devices do not necessitate tooth anchorage, as opposed to fixed appliances with cemented orthodontic bands, which, in certain instances, can potentially lead to infiltrations and cavities on the involved dental elements. Side effects, such as excessive salivation and pain, are mild, and some studies show they gradually disappear with use [32].

### 4.3. Treatment Protocol

The active phase of elastodontic usage is generally 6–8 months with a maximum of one year, with night usage and one or two hours during the day. The second phase is usually characterized by only night usage and lasts 12–16 months. Some myofunctional exercises can be associated with elastodontics to correct bad habits such as oral breathing, atypical swallowing, or lip incompetence.

### 4.4. Treatment Description

Dentoalveolar abnormalities can be treated during pediatric age with elastodontic appliances, either on the sagittal, vertical, or transversal plane. Apart from this use, they eliminate bad habits developed during childhood and stimulate neuromuscular function. Since such appliances do not exert direct forces to generate active orthodontic movement but rather isolate the dentition from muscular forces, treatment outcomes are reached with fewer dentoalveolar compensations [56].

Below, we report a brief description of the clinical fields of applying EAs in the pediatric population, according to the limited available scientific evidence.

#### 4.4.1. Sagittal Discrepancies

The anterior cross bite caused by mild class III or pseudo-class III is one of the reasons why orthodontic interceptive treatment is necessary in order to have correct incisal guidance [38]. In this regard, elastodontic devices, through their vestibular shields, eliminate the centripetal forces generated by the peri-oral musculature and favor a more natural tongue posture that promotes higher transverse and sagittal growth [38]. In fact, in class III subjects with reduced palate dimensions, the tongue is placed in a lower place, and instead of promoting maxillary growth, it stimulates a mandibular one. To break mandibular growth, elastodontics generate posterior pressure in the lower arch [34].

A recent study [38] showed that the use of EAs favors the anteroposterior relationship between the maxillary and mandibular arches with significant skeletal changes (ANB angle, which is the angle formed by point A, nasion, and point B in lateral cephalograms) rather than the simple dentoalveolar changes expected. However, the authors emphasize that skeletal changes are mostly attributed to the decrease in the SNB angle (between point sella, nasion and point B on lateral cephalograms), which, in turn, is influenced by the postural changes of the mandible after the correction of the anterior cross bite. This effect, along with a small increment in the SNA angle (between point sella, nasion, and point A on lateral cephalograms) (not significant) and the vestibular inclination of the upper incisors (significant), can contribute to the interceptive correction of the malocclusion (Figure 2) [38]. Through the mentioned characteristics, negative overjet is corrected.

Another important orthodontic treatment during childhood is the class II malocclusion with high values of OVJ and OVB. In this way, it has been proposed for the use of elastodontic devices (Figure 3).

Through the guided eruption, it is possible to choose how much the posterior teeth can erupt and to apply depressive forces to anterior ones, to reduce overbite values. In class II patients, elastodontics have a mandibular anterior sliding plane to facilitate the advancement of the mandibular jaw, the correction of the overjet via the retroclination of the upper incisors, the proclination of the lower ones the mesial advancement of inferior molars, the opening of the inter-incisal angle and the amelioration of TMJ disorders [34].

Class II malocclusion, from a skeletal point of view, presents high ANB values as follows: with EAs, SNA values detected at pre- and post-treatment at 1 year did not differ significantly from each other, while the pre- and post-treatment recordings of SNB revealed a considerable increase and ANB treatments were reduced to physiological measurements [22,34]. With the lateral cephalometry, a significant increment in the lower anterior [31,40] and total facial height [47] and an increase in mandibular length was observed [31,40,57,58]. Similar results for changes in the cranial and mandibular planes’ relationship between the treated and untreated groups indicate that the EA appliance does not affect the cranio–mandibular vertical relationship [31]. Regarding the morphological change in the mandible, two studies reported a minor tendency for mandibular counterclockwise growth rotation [47,53]. The importance of the correct usage of radiographs in orthodontics, with lateral cephalometry, means that the clinician must evaluate not only his area of work but also the non-working areas. For example, it could be useful to notice the presence of smaller airways, anatomy abnormalities, and the possible presence of a structure such as the pontilicus posticus (PP). This is an abnormal bony prominence of the atlas extending from the posterior portion of the superior articular process to the posterolateral portion of the superior edge of the posterior arch [59]. It creates a passageway for the vertebral artery and the first cervical nerve root [60]. There may be a complete or partial abnormality [59,60]. PP is associated with headaches and neck pain, and it is not a frequent condition. In a recent study, it was reported in 12.6% of patients [61], and for this reason, patients should be informed. It is more frequent in skeletal class II patients and develops frequently between childhood and circumpubertal age [61]; consequently, it should be evaluated first or during orthodontic treatment, as well as with the use of elastodontics.

While several articles [40,47,62] found lingual tipping and retrusion with EAs, a study [57] reported no treatment impact on the inclination or protrusion of the maxillary incisors.

These devices are useful, for all these reasons, to prevent childhood trauma, which is often correlated with higher values of OVJ [22].

In the case of class II patients with an open bite and atypical swallowing, with severe hyper-divergence, the appliance has an elevated occlusal plane in the molar region to obtain the correct vertical dimension [34]. In Figure 4, there is an elastodontic device used in class II patients with atypical swallowing.

Another aspect of class II patients is that, compared to ones with class I and III malocclusions, they have smaller airway volumes [63,64,65,66,67]. To compare different airway volumes, a CBCT can be used, as there are a great amount of software and also a free source to realize the reconstruction of CBCT images for the anatomy. The problem is that the software can underestimate or overestimate the error of the upper airway pattern, and for this reason, for the 3D representation of the upper airway anatomy, it is impossible to expect volumetric consistency between various software programs [68]. Moreover, the use of an automatic deep learning-based method has been proposed to fully automatic the segmentation of the mandible, sinonasal cavity, and pharyngeal airway from CBCTs, with excellent levels of accuracy and a higher speed than an experienced image reader for the first one [69,70].

Returning to small airway volumes of class II patients, two recent studies [32,34] concluded that treatment with elastodontics, in contrast to an untreated control group [32], significantly improves deglutition, phonation, and the respiratory function of upper airways in skeletal class II individuals. In addition, the hyoid bone shifts inferiorly in the treated group compared to the control group at the end of the treatment [32]. For this reason, elastodontics have been proposed to treat OSAS (obstructive sleep apnea syndrome) in children.

#### 4.4.2. Transversal Discrepancies

As reported by different studies [21,22,28,29,34], EAs treat early anterior crowding issues through their design, for example, an elastic channel in the anterior teeth, which stimulates the transversal growth of the dentoalveolar process [39].

Moreover, elastodontic devices can be used to treat functional posterior cross bites. In a recent study [36], IMW (inter-molar width) and ICW (inter-canine width) significantly increased in patients with this device in their mouth, indicating that the EA successfully improved the transverse dimension of the maxillary arch. Interestingly, it has been discovered that the cross bite side sees a higher increase in IMW and ICW than the non-cross bite side, with a substantial decrease in the mean difference between these two sides post-treatment. At the skeleton level, the same thing happened.

In non-severe cases, elastodontics can substitute more invasive therapies, such as rapid maxillary expansion. This latter treatment, in fact, has been associated with external root resorption, even if with a clinically irrelevant value, while the former one does not affect the tooth.

The EA has flanges that isolate the palate and dentoalveolar processes from nearby structures, especially the perioral muscles, which behave asymmetrically in the presence of unilateral posterior cross bites [71,72]. Therefore, it is likely that EAs could help restore a normal pattern of development for the palate and alveolar processes by rebalancing the perioral muscles and contrasting the forces that might otherwise interfere with this growth [36].

#### 4.4.3. Atypical Swallowing, Spoiled Habits, and Teeth Malposition

The Bionator, the Fraenkel, eruption guidance appliances, lingual spurs, fixed grids, speech therapy, and myofunctional treatment are some of the appliances used in modern orthodontics to treat atypical swallowing [37]. Usually, this condition is correlated with the alteration of facial mimicry, the anterior open bite, and oral breathing.

When elastodontic appliances are utilized to move a patient from a compensatory response to a physiological balance, the masticatory muscles respond favorably to treatment [37].

In addition to encouraging muscle development, EAs also prevent bilabial contact, force the tongue into the proper posture, and direct the proper swallowing technique [37].

Moreover, these devices strengthen the back of the tongue, which prevents it from being elevated and positions the tongue’s tip in the proper physiological position [37].

EAs can be used to correct the condition of atypical swallowing in the active phase, either alone or with other treatments, when necessary, but they are also useful during the retention phase when relapse could be possible due to the patient’s bad habits. Atypical swallowing is frequently associated with an open bite in more severe cases, and the use of elastodontics is indicated in pediatric age to improve this condition (Figure 5).

In addition, elastodontics improve breathing (which sometimes is not correct in class II patients, with a tendency to breathe orally), including postural issues and spoiled habits, by eradicating functional stomatognathic system deficiencies and acting in three dimensions within the oral cavity. They can be utilized in conjunction with currently used orthodontic treatments as well as specific functional activities [33].

Despite myofunctional conditions, malocclusions are also correlated with the malposition of teeth, which, as demonstrated in Figure 6, can also be the cause of an anterior cross bite. In this way, elastodontics, through their design in the teeth zone, are used to better align and reestablish physiological occlusal contacts.

#### 4.4.4. Conventional Two-Phase Orthodontics

Interceptive orthodontics has an important role in reducing the possible timing and necessity of fixed treatments. From this point of view, EAs permit the treatment of an extractive case in an interceptive phase, followed by fixed therapy, as reported in a case report [28]. The authors demonstrate that in a 13-year-old patient with class I malocclusion with severe crowding in both arches on a skeletal class II base and maxilla and mandibular arches with severe crowding, augmented overjet (4 mm) and overbite (4.5 mm), the deep curve of Spee and an upper midline shifted to the right, after 4 months of active treatment with EA and myofunctional therapy, had a diminished overbite to an edge-to-edge relationship and the anterior teeth were properly aligned, with some mesial tipping of the first molars [28]. The elastodontic had a design with an anterior teeth channel starting at tooth number 3, an occlusal biteplate, and a temporomandibular shield starting at tooth number 6. Thanks to the elastic anterior teeth channel, the anterior segment on both arches, which was slightly narrower than the average tooth size, could be aligned. The vestibular shield lessened labial and buccal muscle interference, while the biteplate separated the jaws and corrected the increasing overbite [28]. Following EA therapy, fixed treatment became easier and faster for the patient.

#### 4.4.5. Lack of Teeth Retention

One of the advantages of elastodontic devices is minimum anchorage to the teeth, which is particularly useful when a clinician has to treat subjects with dentinogenesis or amelogenesis imperfecta. These diseases implicate a lack of retention for conventional orthodontic therapies, and for this reason, there is a low possibility of improving patient’s oral conditions in mixed dentition [29]. In this regard, computer-aided design/computer-aided manufacturing (CAD–CAM) systems, such as 3D printing technology [73,74], could be used in the upcoming future to generate customized appliances with a full-digital workflow, increasing the precision of these appliances.

### 4.5. Strenghts and Limitations

The study has several strengths, among which is the presence of a meticulously designed and replicable research strategy that involved the examination of all articles on elastodontic devices. Various types of elastodontic devices available in the market were evaluated, and the inclusion of iconographic representations further facilitated comprehension.

For the purpose of clarity, not all of the information reported was well discussed, argued, or presented in this article; some of it was retrieved as a secondary outcome, according to the current systematic narrative hybrid review. In addition, the number of studies included is only 16, and the design of these (often case series or reports) does not permit sufficient supporting literature to define specific conclusions. This literature overview focuses on a small number of studies that dwell on this treatment, so future research should focus on the long-term effects, advantages, and disadvantages of elastodontic devices compared to conventional functional ones to produce scientific and clinical evidence.

## 5. Conclusions

According to the available evidence, elastodontic treatment could represent an effective approach for interceptive orthodontic treatment due to the stimulation and direction of the neuro-musculoskeletal system’s natural growth forces to resolve malocclusions. Moreover, the thermoplastic material makes these appliances comfortable, elastic, and adaptable to any dental arch configuration and is associated with good patient compliance. EAs seem to treat mild-to-moderate dentoskeletal malocclusion in children as well as helpfully prevent bad habits. The present manuscript proposes a literature review of the clinically available evidence on EAs examined in this work, with the aim of familiarizing the reader with the indications and clinical applications of such devices. However, caution should be addressed when considering EAs as an alternative to functional devices due to the absence of solid scientific evidence.

## Figures and Tables

**Figure 1 children-10-01821-f001:**
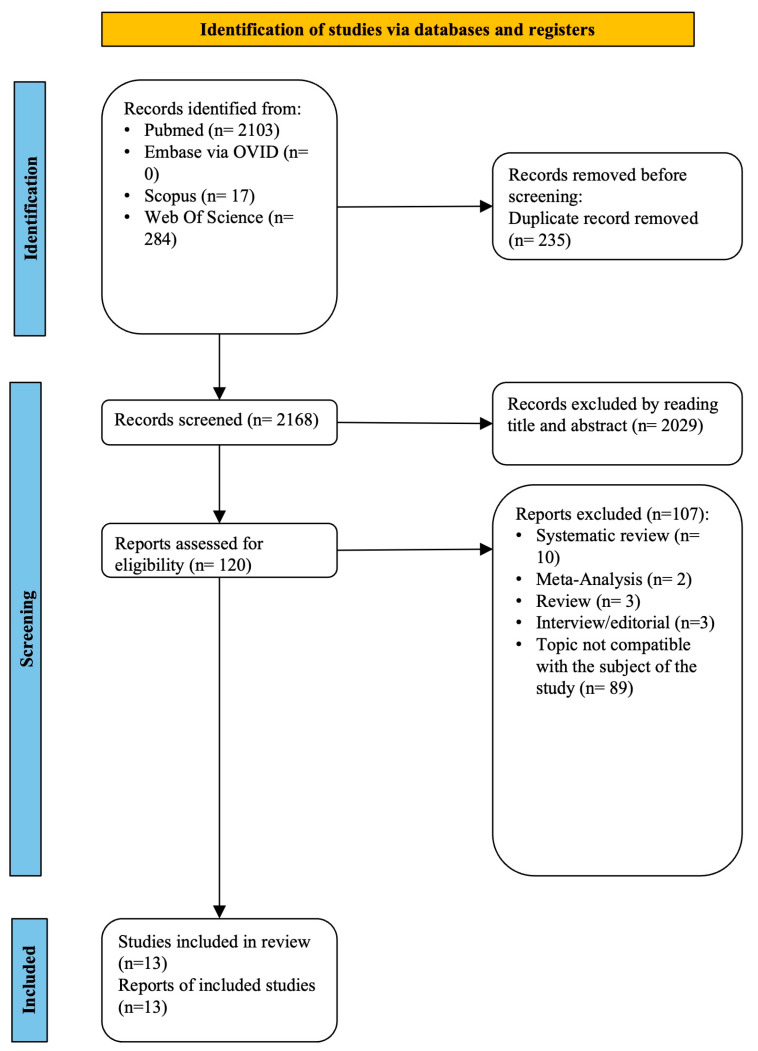
Flow chart of the study selection.

**Figure 2 children-10-01821-f002:**
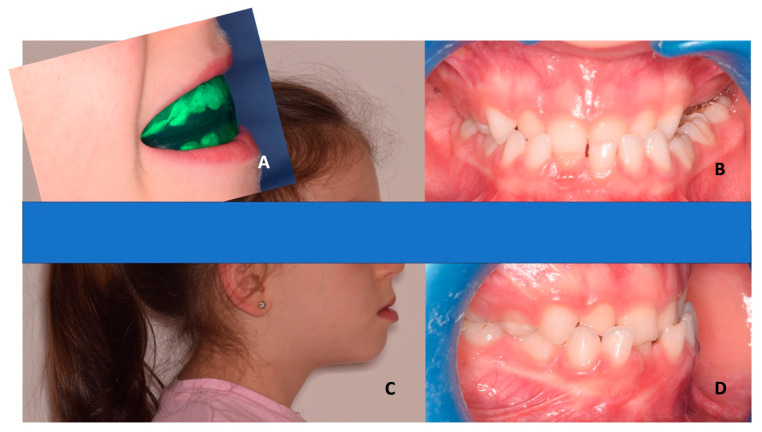
Elastodontic device in a female class III patient (a mean age of 6 years old) with anterior cross bite. (**A**) Lateral view of the patient with the elastodontic device. (**B**) Frontal view of teeth before treatment. (**C**) Profile view before treatment. (**D**) Lateral view of teeth before treatment.

**Figure 3 children-10-01821-f003:**
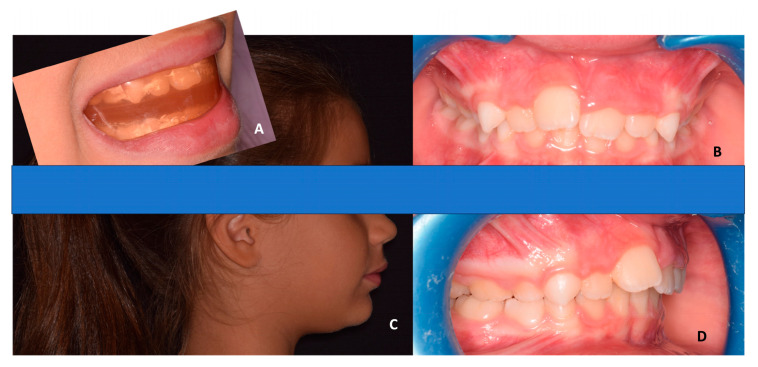
Elastodontic device in a female class II patient (mean age 6 years old) with crowding and high overjet and overbite. (**A**) Lateral view of the patient with the elastodontic device. (**B**) Frontal view of teeth before treatment. (**C**) Profile view before treatment. (**D**) Lateral view of teeth before treatment.

**Figure 4 children-10-01821-f004:**
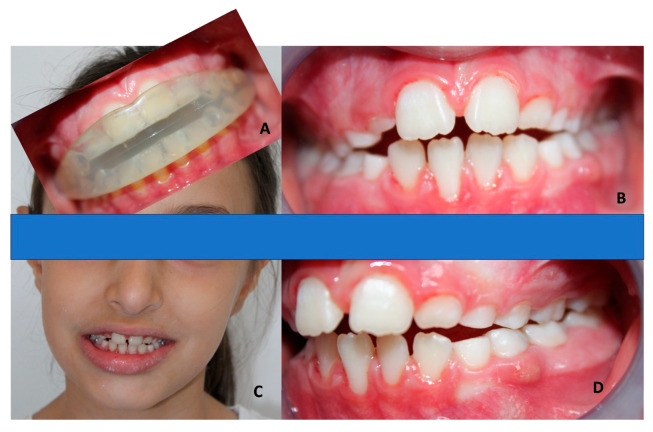
Elastodontic device in a female class II patient (mean age 8 years old) with atypical swallowing. (**A**) Frontal view of the elastodontic device in the mouth. (**B**) Frontal view of the teeth before treatment. (**C**) Frontal view of the patient before treatment. (**D**) Lateral view of the teeth before treatment.

**Figure 5 children-10-01821-f005:**
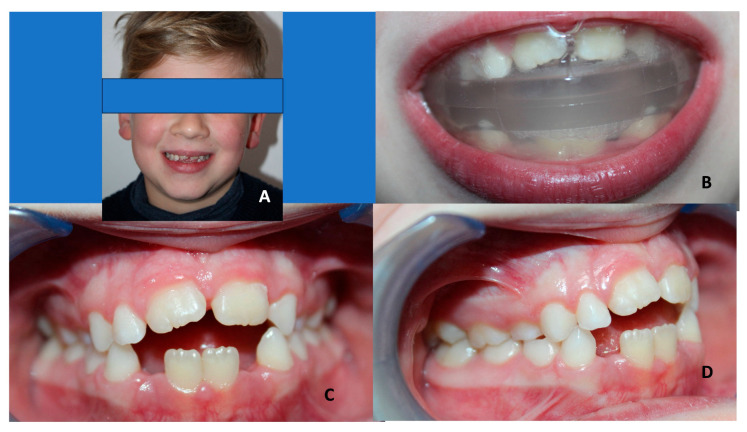
Elastodontic device in a male patient (mean age 7 years old) with atypical swallowing and hyperdivergence. (**A**) Frontal view of the patient before treatment. (**B**) Frontal view of the elastodontic device in the mouth. (**C**) Frontal view of the teeth before treatment. (**D**) Lateral view of the teeth before treatment.

**Figure 6 children-10-01821-f006:**
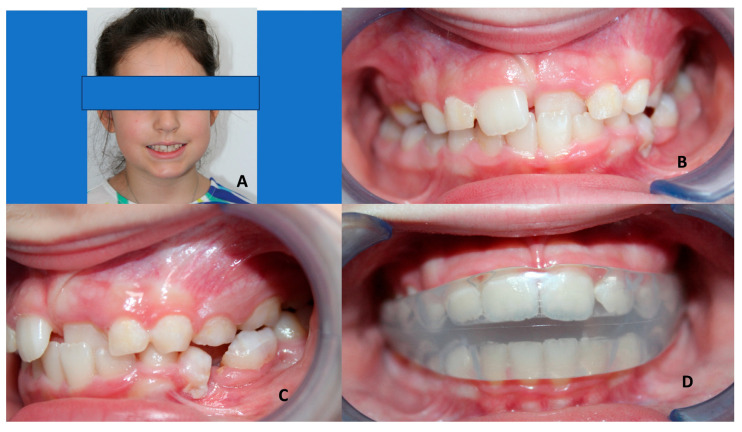
Elastodontic device in a female patient (mean age 8 years old) with teeth malposition. (**A**) Frontal view of the patient before treatment. (**B**) Frontal view of the teeth before treatment. (**C**) Lateral view of the teeth before treatment. (**D**) Frontal view of the elastodontic device in the mouth.

**Table 1 children-10-01821-t001:** Terms used on database search.

Database	Search Format
PUBMED	((“elastodontic”[All Fields] OR (“elastodontic”[All Fields] AND “appliance*”[All Fields]) OR (“elastodontic”[All Fields] AND “device*”[All Fields])) AND ((“orthodontal”[All Fields] OR “orthodontic”[All Fields] OR “orthodontical”[All Fields] OR “orthodontically”[All Fields] OR “orthodontics”[MeSH Terms] OR “orthodontics”[All Fields]) AND (“therapeutics”[MeSH Terms] OR “therapeutics”[All Fields] OR “treatments”[All Fields] OR “therapy”[MeSH Subheading] OR “therapy”[All Fields] OR “treatment”[All Fields] OR “treatment s”[All Fields]))) OR (“orthodontics, interceptive”[MeSH Terms] OR (“orthodontics”[All Fields] AND “interceptive”[All Fields]) OR “interceptive orthodontics”[All Fields] OR (“interceptive”[All Fields] AND “orthodontics”[All Fields]))
EMBASEvia Ovid	(elastodontic) or elastodontic appliance (elastodontic and appliance) or elastodontic device (elastodontic and device) AND orthodontic treatment (orthodontic and treatment) or interceptive orthodontics (interceptive and orthodontics)
WEB OF SCIENCE	((((ALL = (elastodontic)) OR ALL = (elastodontic appliance)) OR ALL = (elastodontic device)) AND ALL = (orthodontic treatment)) OR ALL = (interceptive orthodontics)
SCOPUS	(TITLE-ABS-KEY ((((elastodontic) OR (elastodontic AND appliance)) OR (elastodontic AND device)))) AND TITLE-ABS-KEY (((interceptive AND orthodontics) OR (orthodontic AND treatment)))

## Data Availability

The data presented in this study are available on request from the corresponding author. The data are not publicly available due to to specific ethical and privacy considerations.

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
