# Peer review of "Elastodontic Appliances for the Interception of Malocclusion in Children: A Systematic Narrative Hybrid Review"

_children, 2023, doi:10.3390/children10111821_

Round 1
Reviewer 1 Report
Comments and Suggestions for Authors
Elastodontic Appliances (EAs) for the interception of malocclusion in children: a Narrative Review
Reviewer Report
Thank to the authors for the study. The authors have conducted valuable review study on an important subject. The study, which aims to increase the level of knowledge about the current literature on elastodontic appliances, is important in this respect. However, there are important issues with the presentation and methodology of the review. I think that these issues will improve significantly after being revised carefully and meticulously. Below are my opinions on the study, which requires a major comprehensive revisions:
TITLE:
· Line 2: Abbreviation should not be given in the study title. It should be deleted.
ABSTRACT:
· Results: The number of studies included should be stated. Based on the detailed qualitative analysis results of the studies, it should be stated what the effect(s) of elastodontic appliances are. Accordingly, the most common results should be presented.
· Conclusion: ‘In this paper, current literature about EAs has been analyzed and the possible applications are adequately explained with specific paragraphs, but there is the necessity of future studies, to improve clinical and scientific evidence.’ This statement should be stated in the conclusion section of the main text, not in the conclusion of the abstract section of the study. Therefore it should be deleted. And instead, a conclusion/comment/opinion that can be put forward based on the results from the entire literature review should be stated.
· Line 30: It would be appropriate to add ‘narrative review’ to the keywords.
INTRODUCTION:
· The references in the introduction section of the study are very old. This requires that the study be supported with current reference(s). Therefore, the following statement should be added to line 39: ‘Otherwise, advanced surgical interventions such as orthognathic and/or aesthetic hard or soft tissue surgery are needed to ensure the functional balance between components such as forehead, eyes, chin, nose and lips, which are also important for facial attractiveness in the late period.’ For these statement, the following study should be cited: https://doi.org/10.3390/diagnostics13030463
· Lines 67-70 should be supported by current reference(s).
MATERIALS AND METHODS:
· The information in the abstract of the study should also be stated in more detail in the main text. In this context, the years in which the literature review was conducted, how many people did it, information sources and search strategy, study selection, data collection process and data items, risk of bias, whether the physician who did it was an experienced orthodontist, and quality of evidence assessment should be stated.
· Inclusion and exclusion criteria should be clearly stated.
· Line 104: Use ‘correct’ instead of ‘fix’.
RESULTS:
General comment: The statements under this title of the study do not seem appropriate. In the Results section of a review study, the results of the literature research specified in the material method section should be stated. (As stated above: study selection and characteristics, risk of bias within studies and quality of evidence, results of ındividual studies and data synthesis etc.)
· In this form, the study looks more like a book chapter than a review study. Statements regarding the history, design, clinical use and effects of the appliances should be given under another heading instead of Results. In this context, the Introduction section of the study can be expanded as an Introduction and Background. This is just a sample suggestion. The narrative design can be changed differently, but it should be really changed and revised.
· It would be more accurate to explain the literature results for clinical application under the Discussion heading after lines 150-151. And again, this narrative style should not be like a book chapter.
· It should be stated where the figures are taken from. Additionally, what the different photographs in each figure are should be stated one by one under the figures.
DISCUSSION:
It appears that there is no Discussion section in the study. As mentioned above, it would be appropriate to revise and add it.
CONCLUSIONS:
· Line 319-320: There should not be a statement written with reference in the conclusion section. It is appropriate to state this statement in an appropriate previous section.
REFERENCES:
In general, it seems that there are studies from very old years. For example, 1899, 1980, 1988 etc. In order for a current literature review to provide an up-to-date contribution to the reader, it is important that it is compiled from studies from the last 10-12 years at most. In this sense, those that are 15 years old or earlier studies should be replaced with more up-to-date ones. For this reason, the proposed study should be cited in order to contribute to updating. ( https://doi.org/10.3390/diagnostics13030463 )
Comments on the Quality of English LanguageModerate editing of English language required.
Author Response
Reply to Reviewer # 1
Thank to the authors for the study. The authors have conducted valuable review study on an important subject. The study, which aims to increase the level of knowledge about the current literature on elastodontic appliances, is important in this respect. However, there are important issues with the presentation and methodology of the review. I think that these issues will improve significantly after being revised carefully and meticulously. Below are my opinions on the study, which requires a major comprehensive revisions:
We thank the reviewer for his/her suggestions that have improved the quality of our manuscript.
TITLE:
- Line 2: Abbreviation should not be given in the study title. It should be deleted. According to the reviewer’s request, we’ve made appropriate changes.
ABSTRACT:
- Results: The number of studies included should be stated. Based on the detailed qualitative analysis results of the studies, it should be stated what the effect(s) of elastodontic appliances are. Accordingly, the most common results should be presented. According to the reviewer’s request, we’ve made appropriate changes, reporting in the text “The current literature addressing the usability of EAs is scarce and mostly limited to case reports and case series. After 2168 citations were found through the searches, 16 studies were ultimately included. In this regard, information about the clinical usage and effectiveness of EAs are reported in a narrative form, defining specific domains of application that are clinically oriented (sagittal and transversal discrepancies, atypical swallowing, teeth malposition, two phase orthodontics and lack of teeth retention).” Again, we warmly thank the reviewer for his/her suggestion.
- Conclusion: ‘In this paper, current literature about EAs has been analyzed and the possible applications are adequately explained with specific paragraphs, but there is the necessity of future studies, to improve clinical and scientific evidence.’ This statement should be stated in the conclusion section of the main text, not in the conclusion of the abstract section of the study. Therefore it should be deleted. And instead, a conclusion/comment/opinion that can be put forward based on the results from the entire literature review should be stated. We thank the reviewer for his/her suggestions. we have modified the conclusion in the abstract following his/her guidelines reporting in the text ”Within the intrinsic quality limitation of the available literature, it seems that EAs may represent a promising treatment alternative in managing mild-to-moderate malocclusion in children such as sagittal or transversal discrepancies, crowding, and bad habits such as atypical swallowing.” Again, we warmly thank the reviewer for his/her suggestion.
- Line 30: It would be appropriate to add ‘narrative review’ to the keywords. According to the reviewer’s request, we’ve made appropriate changes
INTRODUCTION:
- The references in the introduction section of the study are very old. This requires that the study be supported with current reference(s). Therefore, the following statement should be added to line 39: ‘Otherwise, advanced surgical interventions such as orthognathic and/or aesthetic hard or soft tissue surgery are needed to ensure the functional balance between components such as forehead, eyes, chin, nose and lips, which are also important for facial attractiveness in the late period.’ For these statement, the following study should be cited: https://doi.org/10.3390/diagnostics13030463 . We thank the reviewer for his/her suggestions. We have added this statement in the text.
- Lines 67-70 should be supported by current reference(s). According to the reviewer’s request, we’ve made appropriate changes
MATERIALS AND METHODS:
- The information in the abstract of the study should also be stated in more detail in the main text. In this context, the years in which the literature review was conducted, how many people did it, information sources and search strategy, study selection, data collection process and data items, risk of bias, whether the physician who did it was an experienced orthodontist, and quality of evidence assessment should be stated.
- Inclusion and exclusion criteria should be clearly stated. According to the reviewer’s request, we’ve made appropriate changes. We have modified this section and explained better the strategy search.
- Line 104: Use ‘correct’ instead of ‘fix’. According to the reviewer’s request, we’ve made appropriate changes
RESULTS:
General comment: The statements under this title of the study do not seem appropriate. In the Results section of a review study, the results of the literature research specified in the material method section should be stated. (As stated above: study selection and characteristics, risk of bias within studies and quality of evidence, results of ındividual studies and data synthesis etc.)
- In this form, the study looks more like a book chapter than a review study. Statements regarding the history, design, clinical use and effects of the appliances should be given under another heading instead of Results. In this context, the Introduction section of the study can be expanded as an Introduction and Background. This is just a sample suggestion. The narrative design can be changed differently, but it should be really changed and revised. We thank the reviewer for his/her suggestions. We have modified the content of results section according to the reviewer’s request.
- It would be more accurate to explain the literature results for clinical application under the Discussion heading after lines 150-151. And again, this narrative style should not be like a book chapter. We thank the reviewer for his/her suggestions. We have added the discussion section according to your suggestion
- It should be stated where the figures are taken from. Additionally, what the different photographs in each figure are should be stated one by one under the figures. According to the reviewer’s request, we’ve made appropriate changes. We reported in the text “Also, discussion of data findings is also supported by iconographic representation of clinical examples of the use of elastodontic appliances in children. Figures were retrieved from the clinical database of the Department of Orthodontics of University of Catania.”
DISCUSSION:
It appears that there is no Discussion section in the study. As mentioned above, it would be appropriate to revise and add it. According to the reviewer’s request, we’ve made appropriate changes. Thanks for this suggestion we added a discussion section.
CONCLUSIONS:
- Line 319-320: There should not be a statement written with reference in the conclusion section. It is appropriate to state this statement in an appropriate previous section. According to the reviewer’s request, we’ve made appropriate changes.
REFERENCES:
In general, it seems that there are studies from very old years. For example, 1899, 1980, 1988 etc. In order for a current literature review to provide an up-to-date contribution to the reader, it is important that it is compiled from studies from the last 10-12 years at most. In this sense, those that are 15 years old or earlier studies should be replaced with more up-to-date ones. For this reason, the proposed study should be cited in order to contribute to updating. ( https://doi.org/10.3390/diagnostics13030463 ). We thank the reviewer for his/her suggestions. The old studies cited in this review are only used to give to the reader a better understanding of the reason of interceptive orthodontics.
Reviewer 2 Report
Comments and Suggestions for Authors
According to this manuscript, I would like to thank the authors for their efforts; it needs revision before evaluating the possibility of publication. I want to pay attention to the following comments:
- The introduction is much summarized, please provide more data about the elastodontic appliances (EAs).
- Please add in detail the age and sex of the patient in the study and the time elapsed from the beginning of the treatment till the end of the treatment.
- Why I did not find the title of "discussion" for the discussion of the results, please rearrange the section of the discussion or merge it with each result.
- Please add more details about the criteria of the included articles (the inclusion and exclusion criteria should be addressed).
- It is preferred to add a flowchart diagram presenting the articles’ selection scheme.
- What is the abbreviation ANB and SNB referred to??.
- The section "5. Conclusions", where is the section "4".
- The cited references are recommended to be within the last 5 years, many references need to be updated as (3, 19, 23 ,34, 50, 57, 58, 62).
- Please add the reference of the paragraph "automatic deep learning-based method for fully automatic segmentation of the mandible from CBCTs, with excellent levels of accuracy and a higher speed than an experienced image reader."
- Please add the limitation of the study at the end of the discussion.
- The conclusions section needs to be rearranged.
Author Response
Reply to Reviewer # 2
We thank the reviewer for his/her suggestions that have improved the quality of our manuscript.
According to this manuscript, I would like to thank the authors for their efforts; it needs revision before evaluating the possibility of publication. I want to pay attention to the following comments:
- The introduction is much summarized, please provide more data about the elastodontic appliances (EAs). We thank the reviewer for his/her suggestions. In the introduction section we give to the reader a general view of elastodontic devices, which is better explained in the discussion section under the heading “design of the device”.
- Please add in detail the age and sex of the patient in the study and the time elapsed from the beginning of the treatment till the end of the treatment. We thank the reviewer for his/her suggestions. Since the study was designed as a review and not as a case-report or case-series, we did not provide clinical information of patients. To satisfy reviewer’s request, we reported these informations in the caption of the figures.
- Why I did not find the title of "discussion" for the discussion of the results, please rearrange the section of the discussion or merge it with each result. We thank the reviewer for his/her suggestions. We have rearranged the text according to the reviewer’s advice.
- Please add more details about the criteria of the included articles (the inclusion and exclusion criteria should be addressed). According to the reviewer’s request, we’ve made appropriate changes and we have explained better the strategy search.
- It is preferred to add a flowchart diagram presenting the articles’ selection scheme. According to the reviewer’s request, we’ve provided the “flow chart” as supplementary material.
- What is the abbreviation ANB and SNB referred to??. We thank the reviewer for his/her suggestions. We have explained what refers to “ANB” and “SNB”.
- The section "5. Conclusions", where is the section "4". We thank the reviewer for his/her suggestions. We have rearranged the text and introduced the discussion section as number four.
- The cited references are recommended to be within the last 5 years, many references need to be updated as (3, 19, 23 ,34, 50, 57, 58, 62). We thank the reviewer for his/her suggestions. The old studies cited in this review are only used to give to the reader a better understanding of the reason of interceptive orthodontics.
- Please add the reference of the paragraph "automatic deep learning-based method for fully automatic segmentation of the mandible from CBCTs, with excellent levels of accuracy and a higher speed than an experienced image reader." We thank the reviewer for his/her suggestions. We had forgotten to insert it. We apologize for the mistake.
- Please add the limitation of the study at the end of the discussion. We thank the reviewer for his/her suggestions. We added it at the end of the discussion.
- The conclusions section needs to be rearranged. We thank the reviewer for his/her suggestions. We have rearranged the conclusion section.
Reviewer 3 Report
Comments and Suggestions for Authors
General Comments:
- Be consistent with your use of terms across the paper e.g. ‘early’ treatment vs ‘interceptive’ treatment
- The English reads okay for the majority of the paper, however, would benefit from proofing by a native English speaker to remove/amend words and phrases that do not fit well into sentences from translation/interpretation services
- The paper cannot be assessed further until the materials and methods section has been completed in accordance with the PRISMA checklist – otherwise the results cannot be verified nor validated and are not scientifically sound at present
- The clinical images used – where have they come from? Permission? Consent? This needs to be mentioned in the informed consent statement.
- In addition, what new information does your paper add, only a few months following publication of Reference 29 - Macrì, M.; Ritrovato, L.; Pisanelli, E.L.; Festa, F. Elastodontic therapy with oral bioactivator devices: A review. Applied Sciences 402 2023, 13, 8868.
o This needs to be explained also
- This paper is not suitable for publication in its current format and needs major revision before being reconsidered – if you cannot provide a more thorough materials and methods, the entire results section cannot be assessed and the paper would need to be rejected.
Specific Comments:
Abstract
- Remove hyphen ‘un-explored’
- Change done to ‘performed’ and give more search data parameters i.e. include from X date until Sept. 2023
- Utilised ‘all discovered keywords and free-standing items’ – this is unclear and requires rewording.
- ‘bad habits’ also needs to be refined or reworded or an example provided.
- The conclusion in the abstract is weak and needs to be entirely reworded.
Introduction
- I do not agree with the use of the term ‘developmental disorder’ – for me ‘anomaly’ or ‘condition’ is a better term
- Again, define or amend the term ‘bad habits’
- The term ‘stomatognathic system’ is not one used in English – please amend.
- L38, I feel you need to add ‘interceptive’ here, and define the age bracket to which you are suggesting i.e. in the early mixed dentition, prior to traditional ages for orthodontic treatment – this goes for L40 also in terms of ‘early treatment’
- ‘basal bones’ – also not a common term – please amend.
- ‘Preschoolers’ – please refine this also, and give age approximations for a global audience to understand.
- Also, how much orthodontic treatment is undertaken in ‘preschoolers’ who would not yet have a mixed-dentition – this requires refinement.
- Line 57/58 – what is this isolated sentence linking to? If the previous one, combine them and include the reference.
- Line 59/60 does not read well – reword and change the term ‘inorganic’.
- Line 61-66 – you need to give some images here of the device both by itself and in the mouth to provide some context – this will improve the paper for the reader
- ‘neuromyofascial system’ – please expand on this for the reader – it is not a common term.
- ‘Fabricated’ I feel has been to literal a translation – Created? Produced? Manufactured?
- Line 73 – available literature is poor – how do you know this prior to performing the review? You mean it ‘appears’ to be poor, or if you knew this already, there would be no need to carry out the review?
Materials and Methods
- The materials and methods section is only eight lines long – this is entirely insufficient and the reliability and validity of the results are therefore entirely in question
- What is the research question?
- Where is the PICO?
- How is anyone supposed to be able to reproduce your searches?
- Where are the numbers of results?
- Where is the flow diagram of results selection and systematic assessment of results?
- Search refinements – MeSH? Boolean operators? Truncation? etc
- Who carried out the results?
- Inclusion/exclusion criteria?
- Who performed abstract and title analysis?
- Was it limited to a specific language?
- The above questions are only SOME of the information is missing and there are many others – Although a narrative review, please provide all the detail listed in the PRISMA checklist – link below
- These are required to make the paper scientifically sound.
Results
- The results cannot be assessed until the materials and methods section has been verified.
Discussion
- You have not provided a discussion section – this is crucial and must be fixed.
- There is no strengths and limitations section
- Implications/ideas for further research needs to be in the discussion.
Conclusions
- You state “although there isn't much available in the way of research, EAs are not susceptible to caries, and they don’t harm the periodontium or promote root resorption” – how do you know this if there isn’t much concrete research?
- The entire conclusion is poorly worded and conflated and must be rewritten.
Comments on the Quality of English LanguageSee comments for authors.
Author Response
Reply to Reviewer # 3
We thank the reviewer for his/her suggestions that have improved the quality of our manuscript.
General Comments:
- Be consistent with your use of terms across the paper e.g. ‘early’ treatment vs ‘interceptive’ treatment. According to the reviewer’s request, we’ve made appropriate changes
- The English reads okay for the majority of the paper, however, would benefit from proofing by a native English speaker to remove/amend words and phrases that do not fit well into sentences from translation/interpretation services We thank the reviewer for his/her suggestions. We agree with the reviewer, the previous version of the manuscript presented several syntax errors. According to this suggestion, we have transmitted the manuscript to a native English speaker.
- The paper cannot be assessed further until the materials and methods section has been completed in accordance with the PRISMA checklist – otherwise the results cannot be verified nor validated and are not scientifically sound at present According to the reviewer’s request, we’ve made appropriate changes. We have rearranged and better explained the materials and methods section.
- The clinical images used – where have they come from? Permission? Consent? This needs to be mentioned in the informed consent statement. We thank the reviewer for his/her suggestions. We added it.
- In addition, what new information does your paper add, only a few months following publication of Reference 29 - Macrì, M.; Ritrovato, L.; Pisanelli, E.L.; Festa, F. Elastodontic therapy with oral bioactivator devices: A review. Applied Sciences 402 2023, 13, 8868.
o This needs to be explained also. We thank the reviewer for his/her suggestions. We know the study mentioned by the reviewer, which was a good study, and, obviously, there are some similaritis between the two studies considering the scarceness of literature on this topic. However, there are also some differences, for example:
- the study of Macrì is focused on a specific type of elastodontic appliance available in the market (AMCOP);
- the study of Macrì provides specific and dip informations about the design of the described appliances (AMCOP);
- our study was based on a more structured and deeper strategy search;
- the wider strategy search allowed us to include different type of elastodontics available today in the market. Indeed, we oriented and addressed our discussion on the general characteristics of elastodontic appliances.
- We also integrate the discussion with iconographic representation of clinical examples of the use of elastodontic appliances in children.
Beside the differences between the two studies, we think that is not necessary to strictly report such differences in the manuscript, but we are willing to do that if the reviewer retains it is necessary.
- This paper is not suitable for publication in its current format and needs major revision before being reconsidered – if you cannot provide a more thorough materials and methods, the entire results section cannot be assessed, and the paper would need to be rejected. We thank the reviewer for his/her suggestions. We have rearranged the material and method section to clarify better the inclusion and exclusion criteria.
Specific Comments:
Abstract
- Remove hyphen ‘un-explored’. According to the reviewer’s request, we’ve made appropriate changes.
- Change done to ‘performed’ and give more search data parameters i.e. include from X date until Sept. 2023. According to the reviewer’s request, we’ve made appropriate changes.
- Utilised ‘all discovered keywords and free-standing items’ – this is unclear and requires rewording. According to the reviewer’s request, we’ve made appropriate changes.
- ‘bad habits’ also needs to be refined or reworded or an example provided. We thank the reviewer for his/her suggestions and we added an example.
- The conclusion in the abstract is weak and needs to be entirely reworded. We thank the reviewer and we have rearranged it.
Introduction
- I do not agree with the use of the term ‘developmental disorder’ – for me ‘anomaly’ or ‘condition’ is a better term. According to the reviewer’s request, we’ve made appropriate changes.
- Again, define or amend the term ‘bad habits’. We thank the reviewer for his/her suggestions and we added an example.
- The term ‘stomatognathic system’ is not one used in English – please amend. We thank the reviewer for his/her suggestions and we modified it.
- L38, I feel you need to add ‘interceptive’ here, and define the age bracket to which you are suggesting i.e. in the early mixed dentition, prior to traditional ages for orthodontic treatment – this goes for L40 also in terms of ‘early treatment’. According to the reviewer’s request, we’ve made appropriate changes.
- ‘basal bones’ – also not a common term – please amend. We thank the reviewer for his/her suggestions and we modified it with the generic term “jaws”
- ‘Preschoolers’ – please refine this also, and give age approximations for a global audience to understand. According to the reviewer’s request, we’ve made appropriate changes.
- Also, how much orthodontic treatment is undertaken in ‘preschoolers’ who would not yet have a mixed-dentition – this requires refinement. According to the reviewer’s request, we’ve made appropriate changes.
- Line 57/58 – what is this isolated sentence linking to? If the previous one, combine them and include the reference. We thank the reviewer for his/her suggestions and we modified it.
- Line 59/60 does not read well – reword and change the term ‘inorganic’. We thank the reviewer for his/her suggestions and we modified it with the generic term “synthetic”.
- Line 61-66 – you need to give some images here of the device both by itself and in the mouth to provide some context – this will improve the paper for the reader. We thank the reviewer for his/her suggestion. Since we have mentioned different type of appliances which can be used for different applications (malocclusion) it is difficult to report images for each type of appliance (we should report a lot of images). However, the discussion section we reported images showing different elastodontic appliances used for different type of malocclusions.
- ‘neuromyofascial system’ – please expand on this for the reader – it is not a common term. We thank the reviewer for his/her suggestions, and we modified it with the generic term “neuomuscular”.
- ‘Fabricated’ I feel has been to literal a translation – Created? Produced? Manufactured? We thank the reviewer for his/her suggestions, and we modified it with the term “manufactured”.
- Line 73 – available literature is poor – how do you know this prior to performing the review? You mean it ‘appears’ to be poor, or if you knew this already, there would be no need to carry out the review? We thank the reviewer for his/her suggestions, and we modified it with the term “appears to be”.
Materials and Methods
- The materials and methods section is only eight lines long – this is entirely insufficient and the reliability and validity of the results are therefore entirely in question We thank the reviewer for his/her suggestions and we have rearranged it to better explain the strategy search.
- What is the research question? We thank the reviewer for his/her suggestions and we added it.
- Where is the PICO? According to the reviewer’s request, we’ve added PICO criteria.
- How is anyone supposed to be able to reproduce your searches? We thank the reviewer for his/her suggestions and we added it.
- Where are the numbers of results? We thank the reviewer for his/her suggestions and we added it.
- Where is the flow diagram of results selection and systematic assessment of results? We thank the reviewer for his/her suggestions and we added it as Figure 1.
- Search refinements – MeSH? Boolean operators? Truncation? Etc We thank the reviewer for his/her suggestions. We added these information on the supplementary Material 1.
- Who carried out the results? According to the reviewer’s request, we’ve made appropriate changes. We added a detailed strategy search in the “Materials and methods”.
- Inclusion/exclusion criteria? We thank the reviewer for his/her suggestions and we added it.
- Who performed abstract and title analysis? According to the reviewer’s request, we’ve made appropriate changes. We added a detailed strategy search in the “Materials and methods”.
- Was it limited to a specific language? We thank the reviewer for his/her suggestions. We reported in the text that it was not limited to a specific language.
- The above questions are only SOME of the information is missing and there are many others – Although a narrative review, please provide all the detail listed in the PRISMA checklist – link below
- These are required to make the paper scientifically sound. We thank the reviewer for his/her suggestions and we have rearranged it to better explain the strategy search following PRISMA checklist.
Results
- The results cannot be assessed until the materials and methods section has been verified. According to the reviewer’s request, we’ve made appropriate changes.
Discussion
- You have not provided a discussion section – this is crucial and must be fixed. According to the reviewer’s request, we’ve made appropriate changes and we have added this section.
- There is no strengths and limitations section. According to the reviewer’s request, we’ve made appropriate changes and we have added this section.
- Implications/ideas for further research needs to be in the discussion. We thank the reviewer for his/her suggestions. We added it.
Conclusions
- You state “although there isn't much available in the way of research, EAs are not susceptible to caries, and they don’t harm the periodontium or promote root resorption” – how do you know this if there isn’t much concrete research? We thank the reviewer for his/her suggestions and we have added the correct reference, we also modified the position of this statement.
- The entire conclusion is poorly worded and conflated and must be rewritten. We thank the reviewer for his/her suggestions and we have rearranged it.
Reviewer 4 Report
Comments and Suggestions for Authors
As a literature review study, this is a meaningful study related to children's orthodontics. However, as a scientific study, minor revisions are necessary as follows:
In the introduction section, the authors clearly indicate the need for evidence-based research and clearly presented the purpose of the study.
In the methods section, although the title of the paper indicates that it is a narrative review, please describe the type of study in the methods. In addition, please add more detailed search procedures for referenced papers (how many papers were referenced, the scope of the period in which the searched papers were published, and the process of organizing the main contents of the referenced papers).
Results section
Lines 170-171, 175-176, 215-216, 272-273, 283-284: Please indicate the literature cited in Figures 1-5.
In the conclusion section, citations should not be used and recommendations should be made based on the key results of this study. Please move the current citation [60] to the ‘Results section’. Additionally, please add the clinical strengths and limitations of this study.

Author Response
Reply to Reviewer # 4
We thank the reviewer for his/her suggestions that have improved the quality of our manuscript.
As a literature review study, this is a meaningful study related to children's orthodontics. However, as a scientific study, minor revisions are necessary as follows:
In the introduction section, the authors clearly indicate the need for evidence-based research and clearly presented the purpose of the study.
In the methods section, although the title of the paper indicates that it is a narrative review, please describe the type of study in the methods. In addition, please add more detailed search procedures for referenced papers (how many papers were referenced, the scope of the period in which the searched papers were published, and the process of organizing the main contents of the referenced papers). We thank the reviewer for his/her suggestions and we have rearranged it to better explain the strategy search.
Results section
Lines 170-171, 175-176, 215-216, 272-273, 283-284: Please indicate the literature cited in Figures 1-5. We thank the reviewer for his/her suggestions. We have taken these photos from the clinical database of the Department of Orthodontics of University of Catania.
In the conclusion section, citations should not be used and recommendations should be made based on the key results of this study. Please move the current citation [60] to the ‘Results section’. Additionally, please add the clinical strengths and limitations of this study. We thank the reviewer for his/her suggestions and we changed the position of the statement. We added the limitations of the study at the end of the discussion.
Round 2
Reviewer 1 Report
Comments and Suggestions for Authors
It has been observed that the review study has improved significantly. It can be published as is.
Comments on the Quality of English LanguageMinor editing of English language required
Author Response
We thank the reviewer for his/her suggestions.
Reviewer 3 Report
Comments and Suggestions for Authors
General Comments:
- I commend the authors for their improved paper and amendments – there are still, however, a number of issues which require improvement, before the paper can be re-assessed for publication.
- You need to be more careful with the wording across the entire paper and report statements as findings rather than fact e.g. ‘side effects….disappear with usage’ – should be ‘side effects such as…. have been shown by some studies to disappear with usage’ – and do this throughout the entire paper.
o E.g. ‘is’ to ‘appears to be’ etc.
- The English still needs largely improving across the paper, there are sentences and adjectives that have been translated and don’t make sense in the dental context e.g. ‘orthopedic devices’ which is not a widely used term in orthodontics in English nor that much of a widely accepted method of treatment
- I believe this paper has potential merit, but needs to be refined further and is currently a little too long.
Specific Comments:
Abstract
- L18 – change ‘up from’ to ‘between July 2023 and September 2023’
- L25-26 – Remove the text from the brackets and add a hyphen before you list the domains
- Don’t list the domains again, you have just mentioned them a few lines above – replace them with the line from your main body conclusion ‘…moderate malocclusion in children, as an adjuvant therapy to the interruption of the spoiled habit’
Introduction
- L37 – ‘oral respiration’ – do you mean ‘mouth breathing’?
- L40 – add ‘has been shown…to be a useful tool’
- L76-85 – You have not put ‘NR’ or ‘SR’, beside any words to show what you abbreviating – amend, as I do not know if ‘SR’ you mean scoping or systematic – please also add references for these definitions/explanations
- L95 – reword to ‘This’ and remove ‘The current’
Materials and Methods
- Reword your research question – you are not looking for ‘guidelines’, nor did your search even include the term ‘guideline’ – maybe, ‘Which malocclusions and/or dental developmental discrepancies may be amenable to treatment with elastodontic devices?” – or something similar.
- L153 – Two writers – but you only give the initials of one?
- The tenses throughout the methods needs to be corrected.
- Please state the specific inclusion and exclusion criteria
- Language limitations?
- Please include the Table of the search terms into the materials and methods section.
Results
- Throughout, please state what level of evidence the information is pertaining to e.g. ‘Two studies, a cohort study and a singular case report, reported…’ – this will allow the reader to assess the strength of the information.
- Tell us the breakdown of the types of studies included e.g. ‘Five were….., four were….’ etc
Discussion
- L303 – Explain what you mean by ‘EAs are not susceptible to caries’ – what has the device got to do with dental caries? You need to expand on this point
- Figure 5 and 6 – Image D needs to be correctly rotated and re-label or reposition them – image ‘D’ should not be the first image.
- 4.5 – Should be ‘Strengths and Limitations’ – State what the benefits of your study are to sell it to the reader – what have you done well? What is unique about your approach/results?
Conclusions
- L512 – ‘orthopedic’? Do you mean orthodontic?
- The majority of the conclusion is not well-written e.g. ‘mild-moderate contraction’ – what is this supposed to mean in regards to orthodontic malocclusion? – Please re-write the conclusion.
- Your conclusion needs to summarise your aims, alongside showing how you met the research question.
Comments on the Quality of English LanguageAs above.
Author Response
We thank the reviewer for his/her suggestions that have improved the quality of our manuscript.
General Comments:
- I commend the authors for their improved paper and amendments – there are still, however, a number of issues which require improvement, before the paper can be re-assessed for publication.
- You need to be more careful with the wording across the entire paper and report statements as findings rather than fact e.g. ‘side effects….disappear with usage’ – should be ‘side effects such as…. have been shown by some studies to disappear with usage’ – and do this throughout the entire paper. According to the reviewer’s request, we’ve made appropriate changes.
o E.g. ‘is’ to ‘appears to be’ etc. According to the reviewer’s request, we’ve made appropriate changes.
- The English still needs largely improving across the paper, there are sentences and adjectives that have been translated and don’t make sense in the dental context e.g. ‘orthopedic devices’ which is not a widely used term in orthodontics in English nor that much of a widely accepted method of treatment According to the reviewer’s request, we’ve made appropriate changes.
- I believe this paper has potential merit, but needs to be refined further and is currently a little too long.
Specific Comments:
Abstract
- L18 – change ‘up from’ to ‘between July 2023 and September 2023’ According to the reviewer’s request, we’ve made appropriate changes.
- L25-26 – Remove the text from the brackets and add a hyphen before you list the domains According to the reviewer’s request, we’ve made appropriate changes.
- Don’t list the domains again, you have just mentioned them a few lines above – replace them with the line from your main body conclusion ‘…moderate malocclusion in children, as an adjuvant therapy to the interruption of the spoiled habit’ According to the reviewer’s request, we’ve made appropriate changes.
Introduction
- L37 – ‘oral respiration’ – do you mean ‘mouth breathing’? We thank the reviewer for his/her suggestions and we have changed it.
- L40 – add ‘has been shown…to be a useful tool’ According to the reviewer’s request, we’ve made appropriate changes.
- L76-85 – You have not put ‘NR’ or ‘SR’, beside any words to show what you abbreviating – amend, as I do not know if ‘SR’ you mean scoping or systematic – please also add references for these definitions/explanations. We thank the reviewer for his/her suggestion and we reported in the text “While scoping reviews are a new tool for mapping the body of research on particular arguments, the two primary techniques for gathering and analyzing literature are narrative reviews (NRs) and systematic reviews (SRs) each one with advantages and disadvantages of their own.”
- L95 – reword to ‘This’ and remove ‘The current’ According to the reviewer’s request, we’ve made appropriate changes.
Materials and Methods
- Reword your research question – you are not looking for ‘guidelines’, nor did your search even include the term ‘guideline’ – maybe, ‘Which malocclusions and/or dental developmental discrepancies may be amenable to treatment with elastodontic devices?” – or something similar. According to the reviewer’s request, we’ve made appropriate changes.
- L153 – Two writers – but you only give the initials of one? We thank the reviewer for his/her suggestion and we reported in the text “Before going through the complete texts of any relevant studies, two writers named V.R. and S.L.R. verified all of the titles and abstracts that they had gathered from the databases.”
- The tenses throughout the methods needs to be corrected. We thank the reviewer for his/her suggestion and we modified the text
- Please state the specific inclusion and exclusion criteria. We thank the reviewer for his/her suggestion and we reported in the text “All the studies concerning elastodontic treatment in interceptive orthodontics were considered, excluding reviews. The studies were also included when they exhibited the following characteristics reported according the PICO format: studies conducted in growing human subjects (Participants); studies evaluating treatment effects using elastodontic devices (Intervention), before and after treatment outcomes or considerations with elastodontic appliances (Comparison), studies assessing occlusal a radiographical data (dental and/or skeletal outcomes) (Outcomes).”
- Language limitations? We thank the reviewer for his/her suggestion and we reported in the text “There were no limitations on the publication year or language.”
- Please include the Table of the search terms into the materials and methods section. According to the reviewer’s request, we reported the table as “Table 1” in the text.
Results
- Throughout, please state what level of evidence the information is pertaining to e.g. ‘Two studies, a cohort study and a singular case report, reported…’ – this will allow the reader to assess the strength of the information. According to the reviewer’s request, we’ve made appropriate changes.
- Tell us the breakdown of the types of studies included e.g. ‘Five were….., four were….’ Etc We thank the reviewer for his/her suggestion and we reported in the text “Three studies were case report [28-30], three were case control [31-33], one was a case series [34], five were retrospective studies [21,35-38], one a prospective study [22].”
Discussion
- L303 – Explain what you mean by ‘EAs are not susceptible to caries’ – what has the device got to do with dental caries? You need to expand on this point We thank the reviewer for his/her suggestion and we reported in the text “From this perspective, it is important to emphasize that elastodontic devices do not necessitate tooth anchorage, as opposed to fixed appliances with cemented orthodontic bands, which, in certain instances, can potentially lead to infiltrations and cavities on the involved dental elements.”
- Figure 5 and 6 – Image D needs to be correctly rotated and re-label or reposition them – image ‘D’ should not be the first image. According to the reviewer’s request, we’ve made appropriate changes.
- 4.5 – Should be ‘Strengths and Limitations’ – State what the benefits of your study are to sell it to the reader – what have you done well? What is unique about your approach/results? We thank the reviewer for his/her suggestion and we reported in the text “The study has several strengths, among which is the presence of a meticulously designed and replicable research strategy that involved the examination of all articles on elastodontic devices. Various types of elastodontic devices available in the market were evaluated, and the inclusion of iconographic representations further facilitates comprehension”
Conclusions
- L512 – ‘orthopedic’? Do you mean orthodontic? According to the reviewer’s request, we’ve made appropriate changes.
- The majority of the conclusion is not well-written e.g. ‘mild-moderate contraction’ – what is this supposed to mean in regards to orthodontic malocclusion? – Please re-write the conclusion. We thank the reviewer for his/her suggestion and we changed the conclusions” According to the available evidence, elastodontic treatment could represent an effective approach for interceptive orthodontic treatment due to the stimulation and direction of neuro-musculoskeletal system's natural growth forces to resolve malocclusions. Moreover, the thermoplastic material makes the appliances comfortable, elastic, and adaptable to any dental arch configuration and is associated with good patients’ compliance. EAs seem to be indicated to treat mild to moderate dento-skeletal malocclusion in children as well as they could be helpful in constrasting bad habits. The present Manuscript proposes a literature review of the clinical available evidence on EAs has been examined in this work, with the aim to familiarize the reader with the indications and clinical applications of such devices. However, caution should be addressed in considering EAs as alternative to functional devices due to the absence of solid scientific evidence.”
- Your conclusion needs to summarise your aims, alongside showing how you met the research question. According to the reviewer’s request, we’ve made appropriate changes.